


# 1 Extending seasonal predictability of Yangtze River summer floods

Shanshan Wang[1,2], and Xing Yuan[1]
[1]Key Laboratory of Regional Climate-Environment for Temperate East Asia (RCE-TEA), Institute of
Atmospheric Physics, Chinese Academy of Sciences, Beijing, 100029, China
[2]Key Laboratory of Arid Climatic Change and Reducing Disaster of Gansu Province, and Key Open
Laboratory of Arid Climate Change and Disaster Reduction of CMA, Institute of Arid Meteorology,
CMA, Lanzhou 730020, China
*Correspondence to*: Xing Yuan (yuanxing@tea.ac.cn)
**Abstract.** Extreme pluvial floods across China's Yangtze River basin in the summer of 2016 was
strongly connected with intense atmospheric moisture transport, and resulted in vast loss of properties
after a strong El Niño winter. Predicting such extreme floods in advance is essential for hazard
mitigation, but the flood forecast skill is relatively low due to the limited predictability of summer
precipitation. By using a "perfect model" assumption, here we show that atmospheric moisture flux has
a higher potential predictability than precipitation over the Yangtze River at seasonal time scales. The
predictability of precipitation and moisture are higher in post-El Niño summers than those in post-La
Niñas, especially for flooding events. As compared with extreme precipitation, the potential
detectability of extreme moisture increases by 20% in post-El Niño summers, which suggests that
atmospheric moisture could be crucial for early warning of Yangtze River summer floods.



## 1. **Introduction**

Located in eastern China with dense population and major agricultural and industrial productions, the Yangtze River basin suffers from frequent flooding due to large interannual variability of the East Asian summer monsoon. In June-July of 2016, extreme pluvial floods hit the middle and lower reaches of the Yangtze River, caused severe inundations over many big cities, and resulted in direct economic loss of 70 billion RMB (about 10 billion U.S. dollars) (Yuan et al., 2018). Effective early warning of upcoming extreme flood events is urgent to mitigate the potential damages, and strongly depends on accurate precipitation forecasts not only at synoptic- but also subseasonal-to-seasonal scales (Yang et al., 2008; Tian et al., 2017). However, predicting flood at seasonal time scales is still a grand challenge due to limited forecast skill in precipitation at long leads (Alfieri et al., 2013; Yuan et al., 2015). This raises the interests to explore other relevant variables that are more predictable than precipitation for flood early warning.

Predictability is an inherent property of the climate system, and it represents the "ability to be predicted" (Boer et al., 2013). As for a numerical prediction model, it is widely accepted that we cannot improve the (precipitation) predictability without improving its dynamical framework, data assimilation and/or physical parameterizations, etc (e.g., Barnston et al., 2012). However, most of the heavy precipitation and flood events in many mid-latitude regions, especially in coastal areas, are strongly related to intense horizontal atmospheric moisture transport (Ralph et al., 2006; Lavers et al., 2014). The atmospheric moisture flux is supposed to be better predicted by large-scale climate models than precipitation that might also be influenced by meso-scale convections (Lavers et al., 2014, 2016b). This provides a potential to use atmospheric moisture flux to extend the predictability of floods. Recently, a series of





studies (Lavers et al., 2014, 2016a, 2016b) have assessed the varying predictability of precipitation and
moisture flux in winter, and shown that moisture flux yields a higher predictability than precipitation at
synoptic-scales (less than two weeks) across northwest Europe and western U.S. that are known as
affected by atmospheric rivers. At sub-seasonal to seasonal time scales, however, whether such
moisture and precipitation predictability relation also applies in China's monsoonal summer seasons
where convection is active, such as the Yangtze River summer flood, is still unclear.
The middle and lower reaches of the Yangtze region River basin in eastern China is one of the most
strongly El Niño-Southern Oscillation (ENSO)-affected regions in the world (e.g., Wang, 2000; Wu et
al., 2003; Ding and Chan, 2005). The persistent Sea Surface Temperature (SST) anomalies in the
equatorial Pacific can alter the tropical and subtropical circulation via local air-sea interaction and/or
teleconnections, and thus affect the East Asia summer climate significantly, including the summer
precipitation in Yangtze region. Such ENSO-related climate anomaly in Yangtze region is not
concurrent with the ENSO cycle, but at a season lag. A possible mechanism for this lag-impact of
ENSO on East Asia summer climate is the Indo-western Pacific ocean capacitor (IPOC), where the
North Indian Ocean warming after El Niño plays a crucial role (Xie et al., 2016). Therefore, the
precipitation predictability over the Yangtze River is closely associated with atmospheric and oceanic
conditions, which is similar to other regions (Gershunov, 1998; Kumar and Hoerling, 1998; Lavers et al.,
2016a). For instance, Kumar and Hoerling (1998) indicated that the North American climate is most
predictable during the late winter and early spring seasons of the warm ESNO events. Lavers et al
(2016a) showed that the moisture flux and extreme precipitation have different predictability during
different North Atlantic Oscillation (NAO) phases. In short, the weather or climate forecasts initialized



at different atmospheric/oceanic conditions can have varying levels of predictability, so understanding
how the Yangtze River rainfall predictability varies during different ENSO phases is also a concern.
In present study, we aim to address the above questions by evaluating the seasonal predictability of
precipitation and moisture flux for the middle and lower reaches of Yangtze River (110-123°E, 27-34°N)
based on multisource observational data, and ensemble hindcasts and real-time forecasts from a
dynamical seasonal forecast model Climate Forecast System version 2 (CFSv2; Saha et al., 2014) for
the period of 1982-2016.
## 2.  Data and Method
### 2.1  Observation and reanalysis data
Monthly mean precipitation data at 1°×1° resolution over Yangtze River basin was obtained from
NOAA's precipitation reconstruction over land (PREC/L), which agrees well with gauge-based datasets
(Chen et al., 2002). Monthly mean atmospheric fields including geopotential height, u-wind, v-wind,
and specific humidity at different pressure levels were derived from the ERA-Interim reanalysis (Dee et
al., 2011). Herein, the mean June-July zonal and meridional atmospheric moisture fluxes between 300
and 1000 hPa were calculated separately, and their magnitudes were combined as the total moisture flux
(Lavers et al., 2016a).
NINO3.4 (5°S–5°N, 120°–170°W) SST anomaly based on ERSSTv4 monthly data (Huang et al., 2014)
during 1948–2016 was used to analyze the impact of ENSO on the seasonal predictability of rainfall
and moisture over the Yangtze River. An ENSO event was defined as the averaged NINO3.4 SST
anomaly during preceding December-January-February (DJF) exceeding its 0.5 standard deviation (σ).
### 2.2 CFSv2 seasonal hindcast and real-time forecast data





The ensemble hindcast and real-time forecast datasets including the monthly specific humidity and wind
field at different levels and monthly precipitation from Climate Forecast System version 2 (CFSv2)
(Saha et al., 2014), were used here to quantify the potential predictability. The predicted moisture flux
was calculated the same as the observation mentioned in Section 2.1. CFSv2 has 24 ensemble members
with different initial conditions (Yuan et al., 2011) and has been widely used for subseasonal to
seasonal forecasting (e.g., Kirtman et al., 2014; Yuan et al., 2015; Tian et al., 2017). All monthly
anomalies were calculated based on the climatology from entire hindcast period (1982-2010). The 0.5-
month lead forecast ensembles started from mid-May to early June (Saha et al., 2014), and predicted
through June-July. Similarly, the 1.5-month lead forecasts for the June-July started from the mid of
April, and so on.
**2.3 The potential predictability approach**
The potential predictability was quantified by using a "perfect model" assumption (Koster et al., 2000,
2004; Luo and Wood, 2006; Becker et al., 2013; Kumar et al., 2014; Lavers et al., 2016b). For the
predictions of June-July mean precipitation and moisture over each grid cell within Yangtze River basin
(110-123°E, 27-34°N) at a given lead time, ensemble member 1 was considered as observation and the
average of members 2–24 was taken as the prediction, which resulted in two time series with 35 years
of record (1982-2016). The skill of this forecast was then calculated by using the anomaly correlation
(AC; Becker et al., 2013) between these two time series, which is defined as $\mathrm{AC} = \dfrac{\sum X'Y'}{[\sum (X')^2 (Y')^2]^{1/2}}$, where
$X'$ is the "observed" precipitation/moisture anomaly and $Y'$ was the predicted counterparts. Here, the 95%
(90%) significant level is 0.33 (0.22) for AC according to a two-tailed Student's t-test. Figure 1 gives an



example of the potential predictability calculation at a grid near Wuhan city, where the ensemble
member 1 was taken as the truth and the mean of the members 2-24 was the prediction. Result shows
that moisture has a higher predictability (AC) than precipitation at 0.5- and 1.5-month lead for member
1. This method was repeated 24 times, with each member being considered as the observation, so as to
obtain 24 AC values; the average of these 24 values was the final estimate of the potential predictability.
In addition to the calculation for individual grid cells, AC value was also calculated by using both
spatial and temporal samples for the Yangtze River basin with 72 CFSv2 grid cells. Here, an AC higher
than 0.05 would be considered as significant at 95% confidence level, both for ENSO events and the
entire period.
The rationale for this "perfect model" approach is that the statistical characteristics of the "observation"
(one of the ensemble members) and the prediction (ensemble mean of remaining members) are the same,
so the estimate of potential predictability is not affected by model biases (Kumar et al., 2014). Generally,
potential predictability is considered as the upper limit of forecasting skill, with an assumption that
internal physics or at least the statistical characteristics of observation and model prediction are
identical (Koster et al., 2004; Kumar et al., 2014).
In addition, the hit rate (HR) was also used to assess the seasonal predictability for extreme hydrologic
events (Ma et al., 2015), where the flooding condition was defined as the June-July mean precipitation
or moisture greater than $90^{th}$ percentile of their climatology. Here, a forecast for flooding event can be
counted at a given grid or region when taking ensemble member 1 as observation and the average of
members 2–24 as the prediction: the HR was computed as $HR = \dfrac{a}{a+c}$, where $a$ represents the number of
events that flooding is forecast and observed, $b$ for flooding is forecast but not observed, and $c$ for



observed flooding that is not forecast. Similar to the AC calculation, 24 HR values would be obtained
when each member was considered as the observation, and their average HR values was the final
potential predictability for extreme hydrologic events.
3. **Results**
**3.1 Yangtze River 2016 pluvial flood and its associated atmospheric circulation**
Figure 2a shows the spatial distribution of the 2016 June-July mean rainfall anomaly. Extreme pluvial
flooding hit the middle and lower reaches of Yangtze River, where the area averaged precipitation
increased by about 40% relatively to the climatology. In particular, continuous heavy rainfall
pummelled the Yangtze River basin, with rainfall anomalies locally exceeding 300m within 10 days
(June 26-July 5; Yuan et al., 2018). Figure 2b shows that the June-July mean precipitation averaged
over the Yangtze River basin ranks only second to the 1954 flood during the period 1948-2016, and is
even heavier than the1998 flood.
This Yangtze River extreme summer flood occurred in the context of the 2015/16 strong El Niño (Zhai
et al., 2016; Yuan et al., 2018). Generally, when the SST over the eastern tropical Pacific is warmer
than normal in the preceding winter, the Yangtze region would experience a wetter summer, or even a
flood hazard. For instance, the catastrophic flooding of the Yangtze River in the summer of 1998 was
strongly influenced by the 1997/98 extreme El Niño (e.g., Lau and Weng, 2001). From November 2015
to January 2016, the seasonal mean SST anomaly in the Niño 3.4 region (NOAA's Oceanic Niño Index)
peaked at 2.3 °C (L'Heureux et al., 2016), and returned to neutral condition until May 2016. With the
influence of the preceding El Niño signal, the western Pacific subtropical high (WPSH) was stronger
than climatology and located further west in the summer of 2016 through the Pacific-East Asian



teleconnection (e.g., Wang, 2000; Wu et al., 2003; Huang et al., 2007; Wang et al., 2014) and the Indo-
western Pacific Ocean capacitor (Xie et al., 2016), so a large amount of moisture was transported along
its western flank, from the Indian ocean, South China Sea and Pacific Ocean to the middle and lower
reaches of Yangtze River (Fig. 2c). As a result, there was a significantly anomalous moisture band in
the east-west direction characterized with the largest moisture transport amount in the middle and lower
reaches of Yangtze River, which was directly responsible for the 2016 summer flood (Fig. 2d).
**3.2 Seasonal predictability of precipitation and moisture flux**
Considering the association between intense moisture flux and heavy rainfall over the Yangtze River
basin, which is known within the canonical East Asian monsoon region (Ding and Chan, 2005), testing
whether atmospheric moisture is more predictable than precipitation at the seasonal time scale is helpful
for flood-control and disaster-relief. Figure 3 shows the predictions for June-July mean anomalies of
precipitation and corresponding moisture flux from the dynamical climate forecast model CFSv2 for the
2016 summer flood at the first three-month leads. As compared with the observed precipitation, CFSv2
successfully captured the rainfall surplus across the middle and lower reaches of the Yangtze River at
0.5-month lead (Fig. 3a), and predicted a visible moisture transport band along the middle and lower
reaches of the Yangtze River (Fig. 3b). The highest moisture anomaly occurred over the southern bank
of Yangtze River, which corresponded exactly to the location of heavy precipitation and flood. At 1.5-
month lead, CFSv2 still performed well for the anomalous moisture flux, but the predicted precipitation
anomaly was much weaker than that at the 0.5-month lead (Figs. 3c-3d). At the 2.5-month lead, the
prediction skill of precipitation significantly weakened with almost no anomaly (Fig. 3e), but the
predicted moisture could reproduce the anomaly to some extent (Fig. 3f).



In addition to the 2016 Yangtze flooding case, potential predictability for June-July precipitation and
moisture flux at different lead times during 1982-2016 is also investigated. Figures 4a-4f depict spatial
distribution of predictability for June-July mean precipitation and moisture flux at the 0.5-, 1.5- and 2.5-
month leads respectively, where moisture flux has higher predictability than precipitation. The highest
AC values for moisture flux occur over the south of the Yangtze River where frequently suffers from
extreme summer pluvial flooding. At the 0.5-month lead, the AC values for precipitation are lower than
0.3 over most areas (Fig. 4a), while they are higher than 0.3 and even close to 0.6 for moisture
predictability over the southern part of the Yangtze River basin (Fig. 4b). The AC value of precipitation
drops quickly over forecast leads, and more than half of the Yangtze region is less than 0.2 when
leading 1.5-month; but the moisture flux performs well with AC values higher than 0.3 and shows good
predictability in the southeast (Figs. 4c-4d). The moisture flux at the 2.5-month lead has higher AC
values even than precipitation at the 0.5-month lead (Fig. 4f). Meanwhile, it is evident that most areas of
the Yangtze River basin have significant predictability (at least at 90% confidence level) for the
moisture flux, but the predictability for precipitation is limited (Figs. 4a-4f).
Figure 4g indicates the corresponding spread for precipitation and moisture predictability throughout
the middle and lower reaches of Yangtze River region (110-123°E, 27-34°N). The median (mean) value
for precipitation is 0.25 (0.23) at the 0.5-month lead, but reaches 0.37 (0.35) for the moisture flux. At
the 2.5-month lead, the median (mean) value for moisture flux is 0.25 (0.24), which is much higher than
the value of 0.18 (0.16) for precipitation. The changes in potential predictability with different forecast
leads are also displayed in Figure 4h, based on both spatial and temporal samples for the Yangtze River
basin. It is evident that moisture flux has consistently higher predictability than precipitation out to 8.5-





month lead. Similar result is also found at the location (30°N, 114°E) near Wuhan city (Fig. 4i), one of
the big cities along the Yangtze River, which suffered widespread inundation in 2016.

**3.3 Varying predictability conditioned on different ENSO phases**

As mentioned above, the Yangtze region in eastern China is one of the most strongly ENSO-affected
regions in the world, and the precipitation variability in this region is generally influenced by the
anomalous ENSO forcing (e.g., Wang, 2000; Wu et al., 2003; Ding and Chan, 2005). To explore their
covariability, here we performed a maximum covariance analysis (MCA, Bretherton et al., 1992) for
preceding December-January-February mean SST (120E-80W, 10S-60N) and June-July mean
precipitation (100E-150E, 10N-55N) fields from 1948 to 2016. It is found that the second mode (MCA2)
explains 23% variance and its corresponding SST anomaly pattern is very similar to the traditional
ENSO-like pattern with a warm anomaly over equatorial eastern Pacific and a horse-shoes cold
anomalies over the western tropical and central Northern Pacific (Fig. 5a). Meanwhile, its temporal
evolution is strongly correlated with the NINO3.4 SST anomaly (r = 0.92, black line in Fig. 5c).
Correspondingly, the summer precipitation in Yangtze region is above normal significantly (Fig. 5b).
Therefore, the Yangtze region is prone to experience a rainy or flooding summer if the SST over the
eastern tropical Pacific is warmer than normal in the preceding winter based on the analysis during the
period 1948-2016, whether the predictability varies during different ENSO phases should be
investigated.
To explore the impacts of preceding El Niño signals on Yangtze precipitation and moisture
predictability, correlations and hit rates conditional on different ENSO phases at different leads are
shown in Figure 6. It is found that the seasonal predictability of Yangtze summer rainfall and moisture





flux is much higher following El Niño years than La Niñas (Fig. 6a). The contrast during different
ENSO phases is more obvious for extreme events, and potential detectability of extreme moisture
increases by 20% in post-El Niño summers as compared with potential detectability of extreme
precipitation (Fig. 6b). Figure 6 also shows that predictability is high conditional on El Niños even out
to 6.5-month lead, which is consistent with previous studies. For instance, Sooraj et al. (2012) have
mentioned that forecasting seasonal rainfall anomalies over central tropical Pacific islands from El Niño
winter into the following spring/summer is skillful by using CFS, and Ma et al. (2015) have
demonstrated high predictability for seasonal drought over ENSO-affected regimes in southern China.
The exception for 3.5-month lead forecast (started in March) where predictability conditioned on La
Niña is slightly higher than El Niño (Fig. 6a) is perhaps related to the 'spring predictability barrier', but
such chaos disappears for extreme events (Fig. 6b).
Furthermore, CFSv2 predictions of atmospheric circulations associated with 500 hPa geopotential
height and 850 hPa wind and moisture flux are also investigated during different ENSO phases. As
shown in Figure 6c, there is an anomalously high pressure center over western subtropical Pacific,
implying that the WPSH is enhanced in post-El Niño summers. Such circulation pattern brings the
moisture from southern oceans to Yangtze River basin, which corresponds well with extreme
hydrologic events. On the contrary, preceding La Niña winters are favorable to a low pressure anomaly
in next summer, accompanied with an abnormal cyclonic circulation, and thereby preventing the
moisture from moving northwards to the Yangtze region (Fig. 6d). The contrast is even obvious at 6.5-
month lead forecasts (Figs. 6e-6f). Such predicted circulation discrepancy in different initial ocean



conditions largely determines higher predictability for extreme hydrologic events over middle and lower
reaches of the Yangtze River basin in post-El Niño summers (Hu et al., 2014).
4. **Summary and Discussion**
Previous studies have revealed that moisture flux has higher predictability than precipitation in weather
forecasts over the northwestern Europe and the western U.S., which is affected by westerlies and
narrow bands of enhanced moisture transport known as atmospheric rivers (Lavers et al., 2014, 2016b).
However, whether the atmospheric moisture is more predictable at seasonal time scales during a
summer monsoon region is still unclear. Based on seasonal ensemble predictions from NCEP's
operational CFSv2 model during 1982-2016, our results show that moisture flux has higher seasonal
predictability than precipitation over China's Yangtze River basin in summer. Moreover, the potential
predictability may change under different climatic conditions. The predictability is much higher when
initialized in warm ENSO conditions not only for precipitation but also for moisture. More importantly,
the moisture shows higher detectability (hit rate) than precipitation for extreme pluvial flooding events
following El Niño winters. The results suggest that it may be possible to extend the predictability of
Yangtze River summer floods and to provide more reliable early warning by using atmospheric
moisture flux predictions. However, to which degree that moisture flux is connected with precipitation
and floods might be model dependent. It is necessary to explore their connections in a multi-model
framework (e.g., NMME; Kirtman et al., 2014; Shukla et al., 2016).
This study extends previous findings on the predictability of precipitation and moisture at synoptic
scales (Lavers et al., 2014) to seasonal time scales, and from atmospheric river-affected regions to the
East Asian summer monsoon region. Given that the transport of atmospheric moisture from oceanic



source regions is important for extreme rainfall in monsoon regions (Gimeno et al., 2012), moisture flux
might also be useful for long-range forecasting over other areas affected by the monsoon and low-level
jets. In fact, extreme precipitation and floods are found to be associated with large-scale moisture
transport over the North American monsoon (Schmitz and Mullen, 1996) and the South American
monsoon (Carvalho et al., 2010) regions.
The higher moisture predictability largely arises from more predictable large-scale circulation (Li et al.,
2016), which strongly determines the moisture transport. Although precipitation variability is affected
by both large-scale moisture transport and localized process and features, such as condensation nuclei in
the atmosphere and lifting movement, it is expected that moisture transport could still be used as a
crucial source of predictability for flooding over monsoonal regimes, especially at long leads where
meso-scale convection is still unpredictable at seasonal time scales.

**Acknowledgement.** This work was supported by National Natural Science Foundation of China
(91547103, 41605055), and the National Key R&D Program of China (2016YFA0600403). The authors
thank Dr. Arun Kumar for helpful discussions. The authors acknowledge NCEP/EMC and IRI
(http://iridl.ldeo.columbia.edu/SOURCES/.NOAA/.NCEP/.EMC/.CFSv2/) for making the CFSv2
hindcast and real-time forecast information available.





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





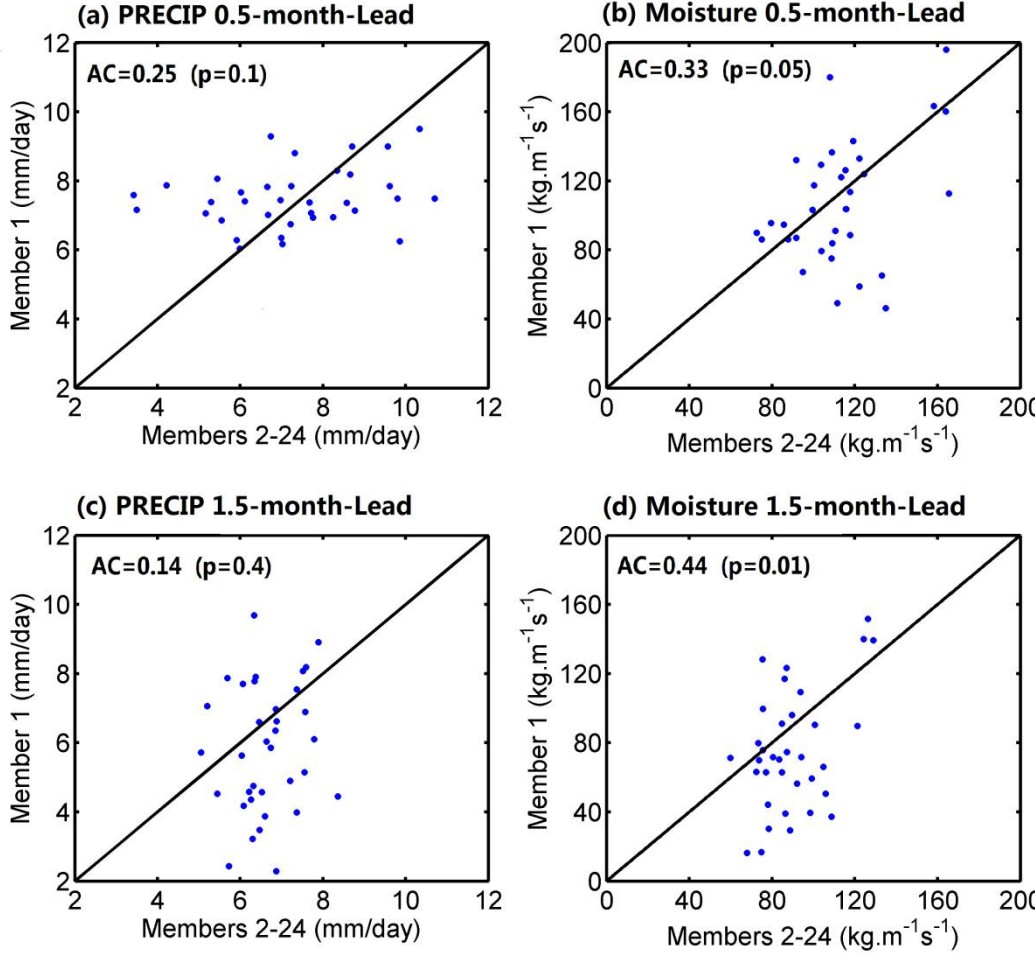

**Figure 1.** An example of the potential predictability calculation, where the ensemble member 1 is the

truth and the mean of the members 2-24 is the prediction. This is for 116°E and 28°N near to Wuhan

city at (a-b) the 0.5-month lead and (c-d) the 1.5-month lead.

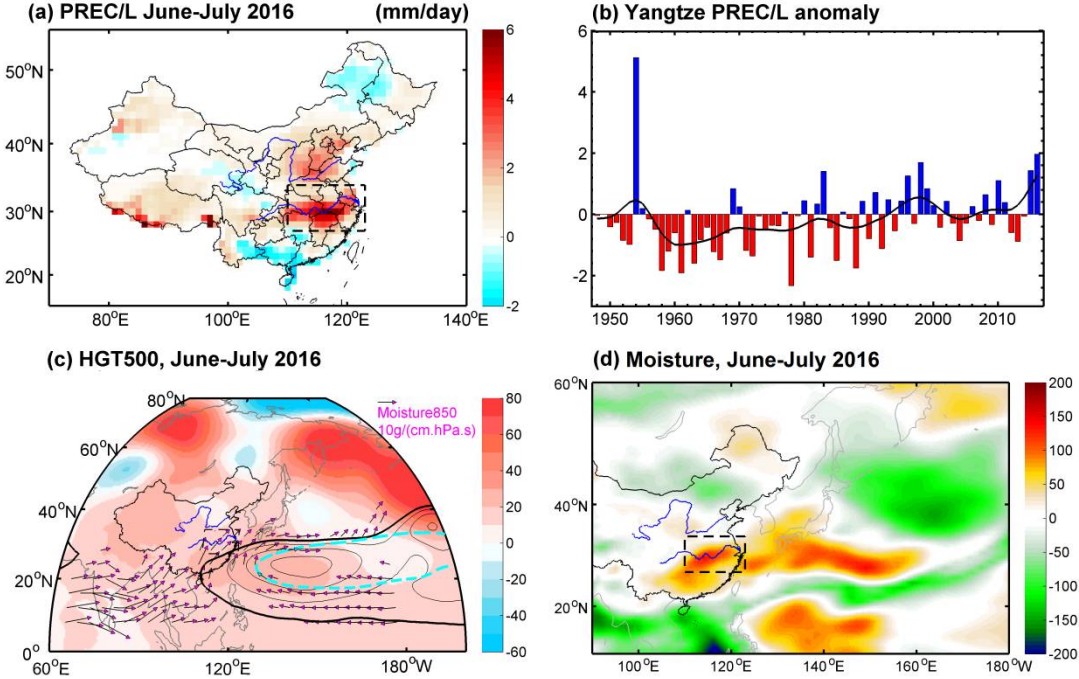

370

**Figure 2.** The 2016 extreme summer flood. (a) Mean precipitation anomaly (shading, mm/day) during the June-July of 2016. (b) Time series of the June-July mean precipitation anomaly averaged over the middle and lower reaches of Yangtze River basin (110-123°E, 27-34°N) in (a). (c) Anomaly of 500 hPa geopotential height (shading, gpm) superimposed by absolute 850 hPa vapor transports (vectors, g/cm•hPa•s). The thick contour lines are 5880 gpm, implying the location of the West Pacific Subtropical High, where the black denotes the June-July 2016 and the cyan is the climatology during 1982-2010. (d) Anomaly of integrated horizontal moisture transport amount between 1000 to 300 hPa layers (shading, Kg•m-1s-1).

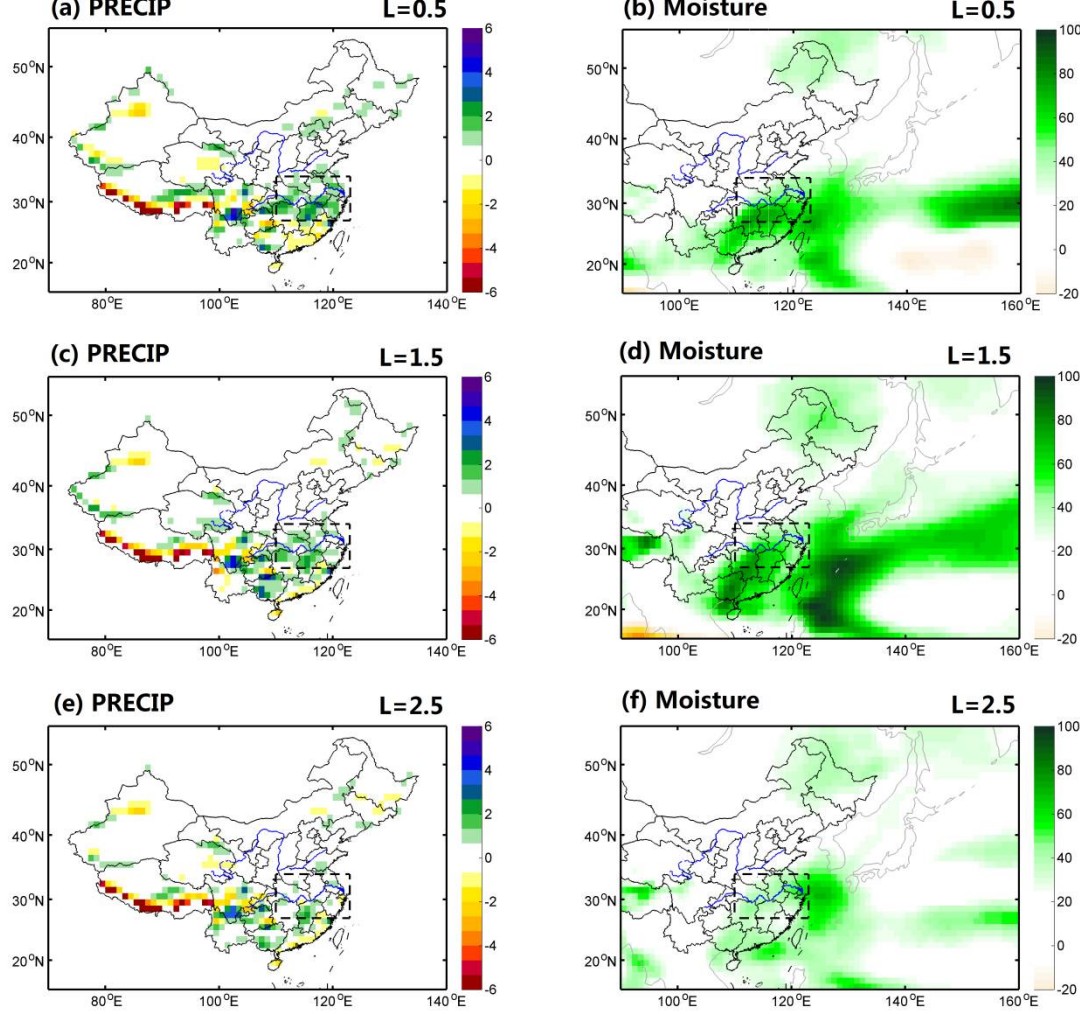

**Figure 3.** Spatial distributions of CFSv2 predicted anomalies of precipitation (shading, mm/day) and atmospheric moisture flux (shading, Kg•m-1s-1) in the June-July of 2016 at the 0.5-, 1.5- and 2.5-month leads.



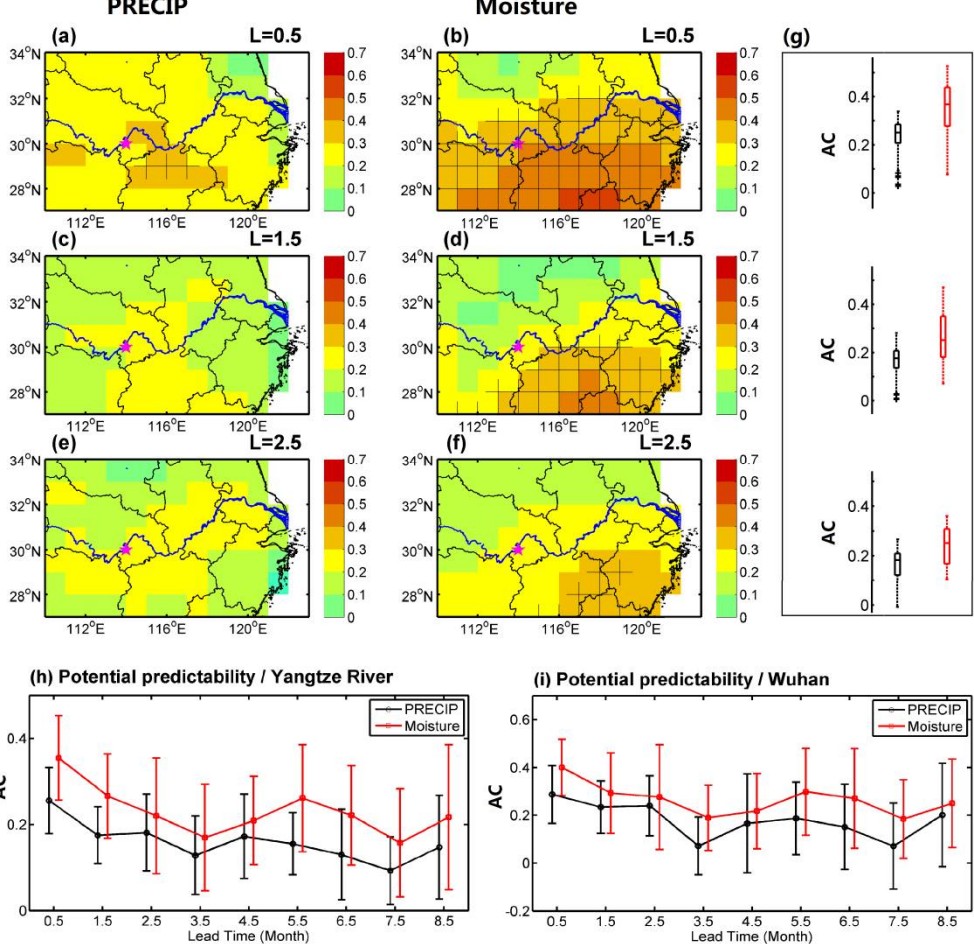

**Figure 4.** (a-f) Potential predictability (AC value, see Method) for June-July mean precipitation and atmospheric moisture flux at different lead times during 1982-2016 over the middle and lower reaches of Yangtze River for the 0.5-, 1.5- and 2.5-month leads; the stippling indicates a 95% confidence level according to a two-tailed Student's t-test. (g) Median, lower and upper quartiles, 1.5 times the interquartile ranges for AC values for precipitation (black) and moisture (red) throughout the study region (110-123°E, 27-34°N); outliers are displayed with + signs. (h-i) Potential predictability throughout study region and Wuhan city (pink pentagram in (a)) at different lead times; the error bars are standard deviations according to 24 members.

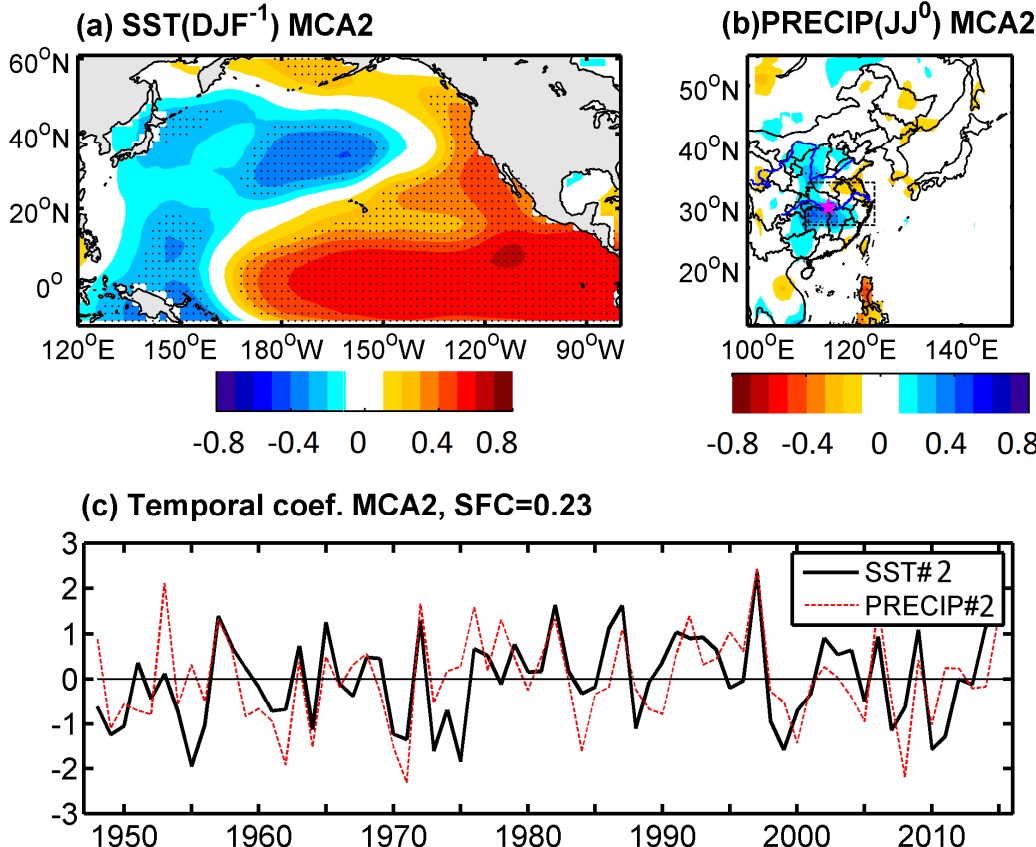

**Figure 5.** (a-b) Spatial and (c) temporal patterns of the second modes based on the maximum covariance analysis (MCA) for SST in preceding winter (December-January-February) and precipitation field in summer (June-July) for 1948-2016. Here the second MCA mode explains 23 % of the variance, as indicated in the square fraction of covariance (SFC).



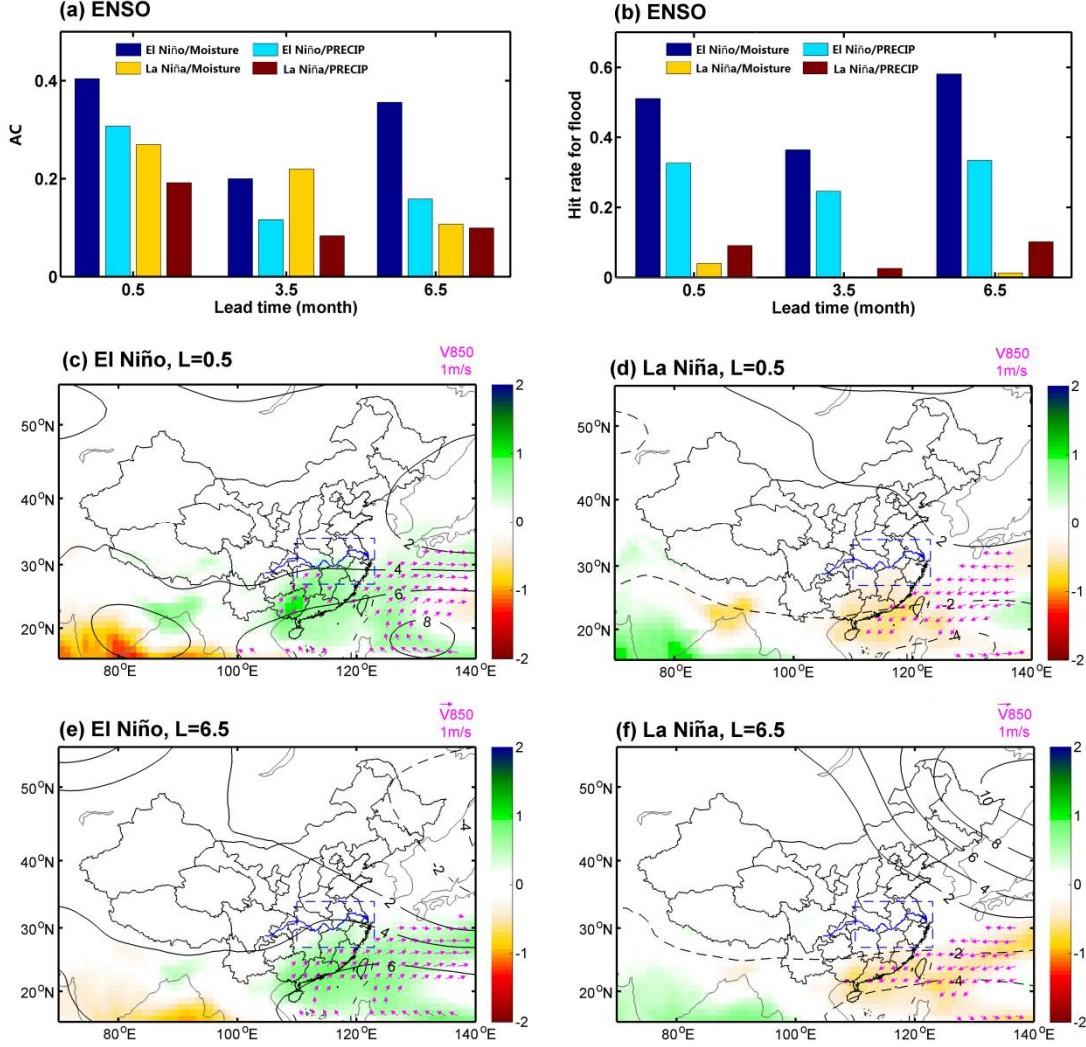

**Figure 6.** Potential predictability at different lead times in terms of (a) anomaly correlation (AC) for

precipitation and moisture, and (b) hit rate (HR) for flood events (>90th percentiles) across the Yangtze

River region conditioned on ENSO phases. (c-d) Composites of predicted anomalies of 500 hPa

geopotential height (contour, gpm) superimposed by 850 hPa wind (vectors, m/s) and moisture flux

(shading, g/cm•hPa•s) at the 0.5-month lead during different ENSO phases. (e-f) The same as (c-d), but

for 6.5-month lead time.