# Peer review of "Extending seasonal predictability of Yangtze River summer floods"

_Hydrology and Earth System Sciences, 2018_

## Referee Comment (RC1) · Anonymous Referee #1 · 10 Apr 2018

This article by Wang and Yuan investigates seasonal predictability of water vapor flux and precipitation in the CFSv2 forecasts to see if water vapor flux could provide higher predictability of floods. I enjoyed reading the study and believe that this is an interesting study and one that is relevant to forecast users. I currently have two major concerns which I lay out below, and some further more minor comments. I hope the authors will find them of use. Major comments:

1. While the language is generally very good, there are grammatical issues, so the paper would benefit from being read by a native English speaker. Also, I think the authors need to be careful with their terminology. For example, on many occasions they say "moisture" when in fact they mean "moisture flux" or "water vapor flux". This needs to be corrected so that readers are not confused by the terms (moisture could

be viewed as total column water vapor).

2. The authors only analyse monthly fields and I believe that this is not sufficient. This is useful, but I think some testing should be done on at least weekly or 2-weekly averaged fields. The reason is that it will be easier (and more beneficial) to see when the predictability drops off in the model at a finer time resolution. For example, Luo and Wood (2006) have some good plots that may help you to consider other time averages.

Other comments:

Line 19: "atmospheric moisture" – see major comment 1.

Line 33: "ability to be predicted" is a strange phrase. Perhaps consider re-phrasing.

Line 40: Precipitation is connected to mesoscale (or more local scale) circulation and orography.

Line 61: I think Lavers et al (2016a) investigated prediction skill, not predictability.

Line 75: What pressure levels were used? Please add some details here.

Lines 103-112: Why does the AC go from 0.33 to 0.44 (compare Figure 1b and 1d)? You would expect the predictability to drop with lead time.

Line 116: I would advise not to use the "upper limit of forecasting skill". This is not always true, as explained in Kumar et al (2014).

Line 119: "Seasonal predictability" is probably seasonal predictive skill.

Line 124: "b" is not in the equation. Please remove.

Line 131: It is hard to see the Yangtze River. Can this figure be edited so that the river is more clear?

Line 133: "pummelled" is not really scientific. Can this be rephrased?

Line 183: Can you consider plotting the differences between precipitation and moisture

flux in Figure 4g? This would more clearly show any significant differences between the variables.

Figure 2: In panel c can you use the total column-integrated moisture flux instead of just the 850 hPa level? This would match the rest of the paper.

Figure 3: What initialisation times are used in this figure (e.g. 1st May 2016)? Please consider adding to the caption.

Figure 6: Panels c-d. Perhaps a few extra contours should be added to more clearly show the 500 hPa geopotential height?

---

## Referee Comment (RC2) · Anonymous Referee #2 · 21 Apr 2018

This study found moisture flux has higher predictability than precipitation in summer in Yangtze River basin, China. The predictability of precipitation and moisture are higher in post-El Niño summers than those in post-La Niñas. The results extend the predictability of Yangtze River summer floods and to provide more reliable early warning by using atmospheric moisture flux predictions.

The research is very interesting and significative.

However, there are a few issues that the authors need to address before the manuscript can be accepted. I recommend most of the issues I raise below just need clarification or justification.

We predict the precipitation in order to predict the flood. How to predict the flood using the moisture? The authors maybe add some discussion.

Line 133, 300m ➔ 300mm.

Line 378, $Kg\cdot m-1s-1$ ➔ $\cdot m^{-1}s^{-1}$

---

## Author Comment (AC1) · 18 May 2018

We are very grateful to the Reviewer for the positive and careful review. The thoughtful comments have helped improve the manuscript. The reviewer's comments are italicized and our responses immediately follow.

**Responses to the comments from Reviewer #1**

This article by Wang and Yuan investigates seasonal predictability of water vapor flux and precipitation in the CFSv2 forecasts to see if water vapor flux could provide higher predictability of floods. I enjoyed reading the study and believe that this is an interesting study and one that is relevant to forecast users. I currently have two major concerns which I lay out below, and some further more minor comments. I hope the authors will find them of use. Major comments:

**Response:** We would like to thank the reviewer for the positive comments. Please see our responses below.

1. While the language is generally very good, there are grammatical issues, so the paper would benefit from being read by a native English speaker. Also, I think the authors need to be careful with their terminology. For example, on many occasions they say "moisture" when in fact they mean "moisture flux" or "water vapor flux". This needs to be corrected so that readers are not confused by the terms (moisture could be viewed as total column water vapor).

**Response:** Thanks for the comments. We have corrected inappropriate terminology and proofread the manuscript carefully. And the "moisture" has been replaced with "moisture flux" throughout the manuscript.

2. The authors only analyse monthly fields and I believe that this is not sufficient. This is useful, but I think some testing should be done on at least weekly or 2-weekly averaged fields. The reason is that it will be easier (and more beneficial) to see when the predictability drops off in the model at a finer time resolution. For example, Luo and Wood (2006) have some good plots that may help you to consider other time averages.

**Response:** Thanks for the comments. According to the reviewer's suggestion, we have incorporated the analysis for the potential predictability of precipitation and moisture flux for weekly averaged fields in the revised manuscript.

We have used the S2S daily data, and have clarified them in the Section "2.2 CFSv2 seasonal hindcast and real-time forecast data" as follows:

"In order to investigate the predictability at finer temporal resolution (e.g., weekly mean fields), the CFSv2 daily reforecasts were obtained from the Subseasonal to Seasonal (S2S) prediction project for the period of 1999-2010, with forecast lead times up to 45-days (Vitart et al. 2017). As for the June 1-7 weekly mean fields, the reforecasts started from May 18 were used as the first ensemble member, the reforecasts started from May 19 were used as the second, and so on. This resulted in 14 ensemble members, with forecast lead times from 1-day to 14-days. The above process was repeated for other weekly averaged fields during June and July. This is called as the first group of ensemble subseasonal forecasts, with lead times of 1-14days. The second group of ensemble reforecasts started from 17 May, 18 May ..., and 30 May were formed similarly, with lead times of 2-15days, and so on."

The AC values for both weekly mean precipitation and moisture flux during June-July at

different lead times were obtained and shown in Figure R1 (Figure 7 in the revised manuscript). Results show that the moisture flux has a higher predictability than precipitation for weekly averaged fields at different lead times, which is consistent with the results on seasonal averaged fields. We have added the related discussion in Section 4 as follows:

"In addition, we also investigated potential predictability of precipitation and moisture flux on weekly averaged fields in June-July at subseasonal scale. Results are similar to seasonal time scale, where the moisture flux has a higher predictability than precipitation at different lead times (Fig. 7)."

Lead time

Figure R1. (a-f) Potential predictability (AC value) for weekly mean precipitation and atmospheric moisture flux at different lead times during June-July of 1999-2010 over the middle and lower reaches of Yangtze River for the 1-14, 5-18 and 8-21 days leads; the stippling indicates a 95% confidence level according to a two-tailed Student's t-test. (g) Potential predictability throughout study region at different lead times. Here, the daily CFSv2 reforecast were obtained from the S2S prediction project for the period of 1999-2010.

Other comments:

*Line 19: "atmospheric moisture" – see major comment 1.* **Response:** Thanks for the suggestion. We have corrected them throughout the manuscript.

*Line 33: "ability to be predicted" is a strange phrase. Perhaps consider re-phrasing.* **Response:** We have changed it to "the ability of the model to predict itself".

Line 40: Precipitation is connected to mesoscale (or more local scale) circulation and orography.

**Response:** Thanks for the comments. We have revised it as follows:

"The atmospheric moisture flux is supposed to be better predicted by large-scale climate models than precipitation that is connected to mesoscale (or more local scale) circulation and orography".

*Line 61: I think Lavers et al (2016a) investigated prediction skill, not predictability.* **Response:** Revised as suggested.

**Line 75: What pressure levels were used? Please add some details here.**

**Response:** We have specified as "Monthly mean atmospheric fields including geopotential height, u-wind, v-wind, and specific humidity at 300, 400, 500, 700, 850, 925 and 1000 hPa were derived from the ERA-Interim reanalysis".

**Lines 103-112: Why does the AC go from 0.33 to 0.44 (compare Figure 1b and 1d)? You would expect the predictability to drop with lead time.**

**Response:** In general, the predictability drops over lead times, but not necessarily for any cases. We plotted the results for all 24 ensemble members in Figure R2, and found that the AC for 0.5-month lead is not necessarily higher than 1.5-month lead. However, the average results for the 24 AC (Fig. R2c) shows that AC decreases over leads on average.

---

## Author Comment (AC2) · 18 May 2018

**Responses to the comments from Reviewer #2**

We are very grateful to the Reviewer for the positive and careful review. The thoughtful comments have helped improve the manuscript. The reviewer's comments are italicized and our responses immediately follow.

This study found moisture flux has higher predictability than precipitation in summer in Yangtze River basin, China. The predictability of precipitation and moisture are higher in post-El Niño summers than those in post-La Niñas. The results extend the predictability of Yangtze River summer floods and to provide more reliable early warning by using atmospheric moisture flux predictions. The research is very interesting and significative. However, there are a few issues that the authors need to address before the manuscript can be accepted. I recommend most of the issues I raise below just need clarification or justification.

**Response:** We would like to thank the reviewer for the positive comments. Please see our responses below.

We predict the precipitation in order to predict the flood. How to predict the flood using the moisture? The authors maybe add some discussion.

**Response:** Thanks for the comments. We have added the discussion as follows:

"Extreme precipitation and floods usually occurred accompanied with intensive atmospheric moisture transport, especially over a large area such as the middle and lower reaches of the Yangtze River. Given higher predictability of atmospheric moisture flux, it can be used as a precursor for flooding forecasting, either directly linking moisture flux to streamflow prediction through statistical techniques (e.g., conditional distribution or Bayesian methods), or adding moisture flux information into precipitation prediction, and consequently improving floods prediction. Moreover, it is suggested that assimilating moisture flux observations into numerical climate forecast models would benefit the prediction of hydrological extremes."

*Line 133, 300m→ 300mm.*

Line 378,  $Kg \bullet m$ -1s-1  $\rightarrow \bullet m^{-1}s^{-1}$

**Response:** Thanks for the comments. We have corrected them as suggested.

---

## Referee Comment (RC3) · Anonymous Referee #3 · 22 May 2018

The article entitled, "Extending seasonal predictability of Yangtze River summer floods" by Wang and Yuan explores the seasonal predictability of both moisture flux and precipitation in the CFSv2 forecast system. The study aims to determine whether moisture flux forecasts can be used to better predict for summer flood prediction (compared to precipitation). I found the study interesting and potentially useful to decision-makers and end-users in the region. However, I have several major concerns that I hope the authors will address, as well as a number of minor comments.

Major comments:

- 1. While much of the study is well written, there are numerous places in the text where there are grammatical issues. These range from simple subject-verb agreement (as in the first sentence, "was" should be replaced with "were"), to passages where the language is misleading and it is not clear what the authors mean to say. The paper (and its corresponding conclusions) would benefit greatly from a thorough proofread by a colleague who can help address and correct the language issues.
- 2. A major conclusion of the study is that the moisture flux can be better predicted than precipitation in summers directly following ENSO events, and particularly El Niño. However, there is very limited discussion of how and why El Niño impacts this area and therefore lends itself as a potential predictor of moisture flux and hence, flooding in the region. Without providing some further discussion to the paragraph that begins on line 220 that speaks directly to how ENSO is understood to impact the area and how the plots shown in Figure 6 are consistent with this, I find that the major conclusions are not fully supported by the study at present. For example, are the moisture flux vectors shown in Figure 6 related to the anomalous high, and is that known to be forced by El Niño? Some more explanation and discussion is needed.

Minor comments:

- 1. Line 39-40, the sentence that mentions model precipitation being influenced by "meso-scale convections" is unclear. Here, are the authors referring to mesoscale (local) circulation patterns that impact precipitation? Also, it might be worth noting that convection schemes themselves (used to parameterize finer scale processes) would also impact forecasted precipitation.
- 2. Line 75: The pressure levels of the variables studied should be identified.
- 3. In Figure 1, is there a reason why the AC is higher for the moisture flux at 1.5 months lead-time compared to 0.5 months? It would be good if the authors could provide some understanding of why this is the case or if they believe it to be spurious because it is surprising.
- 4. Line 124: There is no "b" in the equation on line 123.
- 5. Lines 132-134: This sentence is awkward, particularly the use of the word "pummeled," please rewrite.
- 6. The sentence on Lines 174-177 is also awkward and does not clearly explain the results from Figure 4.
- 7. Line 206: This sentence is a bit contradictory as it says "To explore the impacts of preceding El Nino signals..." and then tells us that "hit rates conditional on

different ENSO phases..." are shown in Figure 6. Figure 6 shows both El Niño and La Niña hit rates, so really the authors are showing the impacts of preceding ENSO events (not just El Niño as is written). Please switch "El Niño" in the beginning of the sentence with "ENSO" and in the second mention of "ENSO" phases, could add "(i.e. El Niño and La Niña)".

- 8. Lines 228-230 conclude that the different circulation patterns predicted for the two ENSO phases determine a higher predictability for extreme hydrologic events in post-El Niño summers. However, why is it necessarily higher predictability and not just a different signal that is predicted because of the different ENSO events? This conclusion seems like a bit of a stretch to me without understanding of why the El Niño signal would translate to higher predictability than La Niña based solely on the evidence presented in the manuscript.
- 9. Line 373 references the "middle and lower reaches of Yangtze River basin." However, these areas are not previously defined in the text. I assume they may be the boxes outlined in Figure 2a, but this needs to be clarified.
- 10. The legend for Figure 2c defines the 850 hPa moisture flux vectors in g/cm\*hPa\*s. I have never seen this unit used before for moisture flux and would recommend it be converted to m/s kg\*kg.
- 11. Figure 3: the different columns are plotted with a different longitudinal domain. It would be helpful in comparing the precipitation to the moisture flux if all panels were plotted using the same longitude bounds.
- 12. Figure 4 seems to contradict what is shown in Figure 1 (see Minor Comment #3). The correlation maps shown in Figure 4 indicate that Wuhan City has a lower AC value for lead-time 1.5 than lead time 0.5, but Figure 1d indicates that the AC is 0.44 for 1.5 month lead but only 0.33 for 0.5 month lead. Why is there a discrepancy?
- 13. While the methods employed are interesting and the figures generally informative, I would encourage some reorganization of Figures 2-6. Figures 2-3 examine the anomalous 2016 event that the text implies is related to the El Niño that occurs that year so when it is followed up by Figure 4 which shows the potential predictability based on all years (1982-2016), it is a bit misleading. I would recommend putting Figure 4 directly after Figure 1 and then continuing on to the Figures detailing the 2015-2016 event.

---

## Author Comment (AC3) · 22 May 2018

We are very grateful to the Reviewer for the positive and careful review. Most of the comments are about clarification and organization, and they are very helpful in improving the manuscript. We will consider them seriously and address them to the extent possible.

Given that the closing date for the open discussion is tomorrow, we will submit our detailed responses to the reviewer after the editor's initial decision for this open discussion.

---

## Author Comment (AC4) · 30 May 2018

**Responses to the comments from Reviewer #3**

We are very grateful to the Reviewer for the positive and careful review. The thoughtful comments have helped improve the manuscript. The reviewer's comments are italicized and our responses immediately follow.

The article entitled, "Extending seasonal predictability of Yangtze River summer floods" by Wang and Yuan explores the seasonal predictability of both moisture flux and precipitation in the CFSv2 forecast system. The study aims to determine whether moisture flux forecasts can be used to better predict for summer flood prediction (compared to precipitation). I found the study interesting and potentially useful to decision-makers and end-users in the region. However, I have several major concerns that I hope the authors will address, as well as a number of minor comments.

**Response:** We would like to thank the reviewer for the positive comments. Please see our responses below.

**Major comments:**

1. While much of the study is well written, there are numerous places in the text where there are grammatical issues. These range from simple subject-verb agreement (as in the first sentence, "was" should be replaced with "were"), to passages where the language is misleading and it is not clear what the authors mean to say. The paper (and its corresponding conclusions) would benefit greatly from a thorough proofread by a colleague who can help address and correct the language issues.

**Response:** Thanks for the comments. We have improved the clarification and carefully proofread the manuscript, including the first sentence.

2. A major conclusion of the study is that the moisture flux can be better predicted than precipitation in summers directly following ENSO events, and particularly El Niño. However, there is very limited discussion of how and why El Niño impacts this area and therefore lends itself as a potential predictor of moisture flux and hence, flooding in the region. Without providing some further discussion to the paragraph that begins on line 220 that speaks directly to how ENSO is understood to impact the area and how the plots shown in Figure 6 are consistent with this, I find that the major conclusions are not fully supported by the study at present. For example, are the moisture flux vectors shown in Figure 6 related to the anomalous high, and is that known to be forced by El Niño? Some more explanation and discussion is needed.

**Response:** Thanks for the comments. We have clarified as follows:

Section 3.3: "As mentioned above, the Yangtze region in eastern China is one of the most strongly ENSO-affected regions in the world, and the precipitation variability in this region is generally influenced by the anomalous ENSO forcing (e.g., Wang, 2000; Wu et al., 2003; Ding and Chan, 2005).....It is found that the second mode (MCA2) explains 23% of the variance, and

its corresponding SST anomaly pattern is very similar to the traditional ENSO-like pattern with a warm anomaly over the equatorial eastern Pacific and a horse-shoes cold anomalies over the western tropical and central Northern Pacific (Fig. 5a). Meanwhile, its temporal evolution is strongly correlated with the NINO3.4 SST anomaly (r = 0.92, black line in Fig. 5c). Correspondingly, the summer precipitation in the Yangtze region is above normal significantly (Fig. 5b)." Above all, there is no doubt that the El Niño signals have an crucial role on the climate variability over the Yangtze region, especially on the precipitation anomalies by impacting the large-scale circulation variation over the Northwestern Pacific Ocean and the associated water vapor transport to the Yangtze region. When El Niño occurs in preceding winter, there is always an enhanced western Pacific subtropical high (WPSH) accompanied with a weakened East Asia summer monsoon (EASM) in the following summer, thereby resulting in an anomalously anticyclonic circulation pattern over the northwestern Pacific that brings large amounts of atmospheric moisture from the oceans to the Yangtze River (Wang et al., 2000; Yuan et al. 2017).

In the revise version, we add some detailed discussion about the mechanism for the lag-impact of El Niño on East Asia summer climate including how the El Niño forcings impact the atmospheric moisture transport to the Yangtze region as follows:

"As shown in Figure 6c, there is an anomalously high pressure center over western subtropical Pacific, which is a recurrent pattern in post-El Niño summers (Xie et al., 2016) and implies that the western Pacific subtropical high (WPSH) is enhanced. Such circulation pattern would bring large amounts of atmospheric moisture from southern oceans to Yangtze River basin, which corresponds well with extreme hydrologic events. The mechanism for this lag-impact of El Niño on East Asia summer climate is the Indo-western Pacific ocean capacitor (IPOC), where the coupled wind–evaporation–SST feedback over Northwest Pacific in spring persists to trigger East Asia–Pacific/Pacific–Japan (EAP/PJ) pattern that arises from the interaction of the anomalous anti-cyclone and North Indian Ocean warming in post-El Niño summers (Xie et al., 2016)." (L241-250 in the tracked version of the revised manuscript)

**Minor comments:**

1. Line 39-40, the sentence that mentions model precipitation being influenced by "meso-scale convections" is unclear. Here, are the authors referring to mesoscale(local) circulation patterns that impact precipitation? Also, it might be worth noting that convection schemes themselves (used to parameterize finer scale processes) would also impact forecasted precipitation.

**Response:** Thanks for the comments. We have revised the manuscript as follows:

"The atmospheric moisture flux is supposed to be better predicted by large-scale climate models than precipitation that is not only connected to mesoscale (or more local scale) circulation but also influenced by the vertical convection and localized orography (Lavers et al., 2014, 2016b)." (L39-42)

2. Line 75: The pressure levels of the variables studied should be identified.

**Response:** Thanks for the comments. We have specified as "Monthly mean atmospheric fields including geopotential height, u-wind, v-wind, and specific humidity at 300, 400, 500, 700, 850, 925 and 1000 hPa were derived from the ERA-Interim reanalysis". (L77-79)

3. In Figure 1, is there a reason why the AC is higher for the moisture flux at 1.5months lead-time compared to 0.5 months? It would be good if the authors could provide some understanding of why this is the case or if they believe it to be spurious because it is surprising. **Response:** Thanks for the comments. In general, the predictability drops over lead times, but not

necessarily for any cases.

We plotted the results for all 24 ensemble members in Figure R1, and found that the AC for 0.5-month lead is not necessarily higher than 1.5-month lead. However, the average results for the 24 AC (Fig. R1c) shows that AC decreases over leads on average.

**Figure R1**. Potential predictability (AC value) when different ensemble member was taken as the truth and the mean of the members was the prediction at Wuhan city for the (a) 0.5-and (b) 1.5-month leads. (c) the final estimate of the potential predictability in Wuhan city.

4. Line 124: There is no "b" in the equation on line 123.

Response: We have removed it.

5. Lines 132-134: This sentence is awkward, particularly the use of the word "pummeled," please rewrite.

**Response:** Thanks for the comments. We have changed it as "In particular, continuous heavy rainfall hit the Yangtze River basin, with rainfall anomalies locally exceeding 300 mm within 10 days (June 26-July 5; Yuan et al., 2018)". (L145-147)

**6. The sentence on Lines 174-177 is also awkward and does not clearly explain the results from Figure 4.**

**Response:** Thanks for the comments. We have revised as "The AC values for precipitation drop quickly with forecast leads, and Fig. 4c shows that more than half of the AC values are less than 0.2 over the Yangtze region at 1.5-month lead. However, the moisture flux performs well with

many AC values higher than 0.3 at 1.5-month lead, especially over the south eastern mountain region (Fig. 4d)." (L187-191)

7. Line 206: This sentence is a bit contradictory as it says "To explore the impacts of preceding El Nino signals..." and then tells us that "hit rates conditional on different ENSO phases..." are shown in Figure 6. Figure 6 shows both El Niño and La Niña hit rates, so really the authors are showing the impacts of preceding ENSO events (not just El Niño as is written). Please switch "El Niño" in the beginning of the sentence with "ENSO" and in the second mention of "ENSO" phases, could add "(i.e. El Niño and La Niña)".

**Response:** Thanks for the comments. We have revised as suggested.

"To explore the impacts of preceding ENSO signals on Yangtze precipitation and moisture flux predictability, correlations and hit rates conditional on different ENSO phases (i.e., El Niño and La Niña) at different leads are shown in Figure 6." (L222-224)

8. Lines 228-230 conclude that the different circulation patterns predicted for the two ENSO phases determine a higher predictability for extreme hydrologic events in post-El Niño summers. However, why is it necessarily higher predictability and not just a different signal that is predicted because of the different ENSO events? This conclusion seems like a bit of a stretch to me without understanding of why the El Niño signal would translate to higher predictability than La Niña based solely on the evidence presented in the manuscript.

**Response:** Thanks for the comments. We have added more explanations in the revised manuscript as follows:

"The asymmetric performance during El Niño and La Niña has drawn many attentions. One of the reasons is that the atmospheric response to tropical Pacific SST anomaly is inherently nonlinear (Hoerling et al., 1997), where both the amplitude of SST anomaly in the eastern equatorial Pacific and the associated atmospheric response are significantly larger during El Niño than during La Niña episodes (Burgers and Stephenson 1999)." (L228-232)

"It implies that the precipitation deficits or droughts are more likely to occur in this region in post-LaNiña summers. The contrast is obvious even for forecasts at 6.5-month lead (Figs. 6e-6f). The differences in predicted circulation and associated moisture transport largely result in higher predictability for extreme hydrologic events over middle and lower reaches of the Yangtze River basin in post-El Niño summers (Hu et al., 2014)." (253-258)

9. Line 373 references the "middle and lower reaches of Yangtze River basin."However, these areas are not previously defined in the text. I assume they may be the boxes outlined in Figure 2a, but this needs to be clarified.

**Response:** We have now defined it in the Introduction section as follows:

"In present study, we aim to address the above questions by evaluating the seasonal predictability of precipitation and moisture flux for the middle and lower reaches of Yangtze River (110-123°E, 27-34°N) based on multisource observational data, and ensemble hindcasts

and real-time forecasts from a dynamical seasonal forecast model Climate Forecast System version 2 (CFSv2; Saha et al., 2014) for the period of 1982-2016." (L68-72)

10. The legend for Figure 2c defines the 850 hPa moisture flux vectors in g/cm\*hPa\*s. I have never seen this unit used before for moisture flux and would recommend it be converted to m/s kg\*kg.

**Response:** Thanks for the comments. According to the suggestion from reviewer#1, we have used the total column-integrated moisture flux instead of that at the 850 hPa level in revised manuscript. The corresponding unit has also been converted to kg•m-1s-1. (L418-426)

---

## Author Response (AR1)

**1 Extending seasonal predictability of Yangtze River summer floods**

2 Shanshan Wang1, 2, and Xing Yuan1

[revised manuscript text omitted]

97 In order to investigate the predictability at finer temporal resolution (e.g., weekly mean fields), the 98 CFSv2 daily reforecasts were also obtained from the Subseasonal to Seasonal (S2S) prediction project 99 for the period of 1999-2010, with the forecast lead times up to 45-days (Vitart et al. 2017). As for the 90 June 1-7 weekly mean fields, the reforecasts started from May 18 were used as the first ensemble 91 member, the reforecasts started from May 19 were used as the second, and so on. This resulted in 14 92 ensemble members, with forecast lead times from 1-day to 14-days. The above process was repeated for 93 other weekly averaged fields during June and July. This is called as the first group of ensemble subseasonal forecasts, with lead times of 1-14days. The second group of ensemble reforecasts started

105 from 17 May, 18 May ..., and 30 May were formed similarly, with lead times of 2-15 days, and so on.

**106 2.3 The potential predictability approach**

The potential predictability was quantified by using a "perfect model" assumption (Koster et al., 2000, 2004; Luo and Wood, 2006; Becker et al., 2013; Kumar et al., 2014; Lavers et al., 2016b). For the predictions of June-July mean precipitation and moisture flux over each grid cell within the Yangtze River basin (110°-123°E, 27°-34°N) at a given lead time, ensemble member 1 was considered as the observation and the average of members 2–24 was taken as the prediction, which resulted in two time series with 35 years of record (1982-2016). The skill of this forecast was then calculated by using the anomaly correlation (AC; Becker et al., 2013) between these two time series, which is defined as

114
$$\underline{AC} = \frac{\sum X' Y'}{\left[\sum (X')^2 (Y')^2\right]^{1/2}} \underline{AC} = \frac{\sum X' Y'}{\left[\sum (X')^2 (Y')^2\right]^{1/2}}, \text{ where } X' \text{ is the "observed" precipitation/moisture flux}$$

[revised manuscript text omitted]
 values of 188 precipitation drops quickly with over forecast leads, and Fig. 4c shows that more than half of the 189 Yangtze region more than half of the AC values areis less than 0.2 over- the Yangtze region at the 190 when leading-1.5-month lead.; However, but the moisture flux still performs well with many AC values 191 higher than 0.3 at the 1.5-month lead, especially over and shows good predictability in the southeastern 192 193 mountain region (Fig. s. 4e-4d). The moisture flux at the 2.5-month lead has higher AC values even than precipitation at the 0.5-month lead (Fig. 4f). Meanwhile, it is evident that most areas of the 194 Yangtze River basin have significant predictability (at least at 90% confidence level) for the moisture 195 flux, but the predictability for precipitation is limited (Figs. 4a-4f). 196

197 Figure 4g indicates the corresponding spread for precipitation and moisture flux predictability throughout the middle and lower reaches of Yangtze River region (110°-123°E, 27°-34°N). The median 198 199 (mean) value for precipitation is 0.25 (0.23) at the 0.5-month lead, but reaches 0.37 (0.35) for the moisture flux. At the 2.5-month lead, the median (mean) value for moisture flux is 0.25 (0.24), which is 200 201 much higher than the value of 0.18 (0.16) for precipitation. The changes in potential predictability with 202 different forecast leads are also displayed in Figure 4h, based on both spatial and temporal samples for the Yangtze River basin. The difference between precipitation and moisture flux is statistically 203 significant (p < 0.05) with a two-tailed Student's t-test. It is evident that moisture flux has consistently 204 higher predictability than precipitation out to 8.5-month lead. Similar result is also found at the location 205

206 (30°N, 114°E) near Wuhan city (Fig. 4i), one of the big cities along the Yangtze River, which suffered
207 widespread inundation in the summer of 2016.

**208 3.3 Varying predictability conditioned on different ENSO phases**

As mentioned above, the Yangtze region in eastern China is one of the most strongly ENSO-affected 209 regions in the world, and the precipitation variability in this region is generally influenced by the 210 anomalous ENSO forcing (e.g., Wang, 2000; Wu et al., 2003; Ding and Chan, 2005). To explore their 211 covariability, here we performed a maximum covariance analysis (MCA, Bretherton et al., 1992) for the 212 preceding December-January-February mean SST (120°E-80°W, 10°S-60°N) and June-July mean 213 214 precipitation (100°E-150°E, 10°N-55°N) fields from 1948 to 2016. It is found that the second mode (MCA2) explains 23% of the variance, and its corresponding SST anomaly pattern is very similar to the 215 traditional ENSO-like pattern with a warm anomaly over the equatorial eastern Pacific and a horse-216 shoes pattern with cold anomalies over the western tropical and central nNorthern Pacific (Fig. 5a). 217 Meanwhile, its temporal evolution is strongly correlated with the NINO3.4 SST anomaly (r = 0.92, 218 black line in Fig. 5c). Correspondingly, the summer precipitation in the Yangtze region is above normal 219 significantly (Fig. 5b). Therefore, the Yangtze region is prone to experience a rainy or flooding summer 220 if the SST over the eastern tropical Pacific is warmer than normal in the preceding winter based on the 221 222 covariance analysis during the period 1948-2016, whether the predictability varies during different 223 ENSO phases should be investigated.

To explore the impacts of preceding ENSO El Niño signals on Yangtze precipitation and moisture flux
predictability, correlations and hit rates conditional on different ENSO phases (i.e., El Niño and La Niña)
at different leads are shown in Figure 6. It is found that the seasonal predictability of Yangtze summer

| 227 | rainfall and moisture flux is much higher following El Niño years than La Niñas (Fig. 6a). The contrast    |
|-----|------------------------------------------------------------------------------------------------------------|
| 228 | during different ENSO phases is more obvious for extreme events, and the potential detectability of        |
| 229 | extreme moisture flux increases by 20% in post-El Niño summers as compared with the potential              |
| 230 | detectability of extreme precipitation (Fig. 6b). This asymmetric performance during El Niño and La        |
| 231 | Niña has drawn many attentions. One of the reasons is that the atmospheric response to tropical Pacific    |
| 232 | SST anomaly is inherently nonlinear (Hoerling et al., 1997), where both the amplitude of SST anomaly       |
| 233 | in the equatorial eastern Pacific and the associated atmospheric response are significantly larger during  |
| 234 | El Niño than during La Niña episodes (Burgers and Stephenson 1999). Figure 6 also shows that the           |
| 235 | predictability is high conditional on El Niños even out to 6.5-month lead, which is consistent with        |
| 236 | previous studies. For instance, Sooraj et al. (2012) have mentioned that forecasting seasonal rainfall     |
| 237 | anomalies over central tropical Pacific islands from El Niño winter into the following spring/summer is    |
| 238 | skillful by using CFS, and Ma et al. (2015) have demonstrated high predictability for seasonal drought     |
| 239 | over ENSO-affected regimes in southern China. The exception for 3.5-month lead forecast (started in        |
| 240 | March) where the predictability conditioned on La Niña is slightly higher than El Niño (Fig. 6a) is        |
| 241 | perhaps related to the 'spring predictability barrier', but such chaos disappears for extreme events (Fig. |
| 242 | 6b),                                                                                                       |
| 243 | Furthermore, CFSv2 predictions of atmospheric circulations associated with 500 hPa geopotential            |
| 244 | height and 850 hPa wind and moisture flux are also investigated during different ENSO phases. As           |
| 245 | shown in Figure 6c, there is an anomalously high pressure center over the subtropical western              |

247 impliesying that the WPSH is enhanced in post El Niño summers. Such circulation pattern would brings

246

subtropical Pacific, which is a recurrent pattern in post-El Niño summers (Xie et al., 2016) and

the larger amounts of atmospheric moisture than normal from the southern oceans to the Yangtze River 248 basin, which corresponds well with extreme hydrologic events. The mechanism for this lag-impact of El 249 Niño on East Asia summer climate is the Indo-western Pacific ocean capacitor (IPOC), where the 250 coupled wind-evaporation-SST feedback over the Northwest Pacific in spring persists to trigger East 251 Asia-Pacific/Pacific-Japan (EAP/PJ) pattern that arises from the interaction of the anomalous anti-252 cyclone and North Indian Ocean warming in post-El Niño summers (Xie et al., 2016). On the contrary, 253 preceding La Niña winters are favorable to a low pressure anomaly in next summer, accompanied with 254 an abnormal cyclonic circulation, and thereby preventing the moisture from moving northwards to the 255 256 Yangtze region (Fig. 6d). It implies that the precipitation deficits or droughts are more likely to occur in this region in post-La Niña summers. The contrast is even obvious even for forecasts forat 6.5-month 257 lead forecasts (Figs. 6e-6f).- The differences in predicted circulation and associated moisture transport 258 largely result in Such predicted circulation discrepancy in different initial ocean conditions largely 259 260 determines higher predictability for extreme hydrologic events over the middle and lower reaches of the Yangtze River basin in post-El Niño summers (Hu et al., 2014). 261

**262 4. Summary and Discussion**

Previous studies have revealed that moisture flux has higher predictability than precipitation in weather forecasts over the northwestern Europe and the western U.S., which areis affected by westerlies and narrow bands of enhanced moisture transport known as atmospheric rivers (Lavers et al., 2014, 2016b). However, whether the atmospheric moisture flux is more predictable at seasonal time scales during a summer monsoon region is still unclear. Based on seasonal ensemble predictions from NCEP's operational CFSv2 model during 1982-2016, our results show that moisture flux has higher seasonal

predictability than precipitation over China's Yangtze River basin in summer. In addition, we also 269 investigated potential predictability of precipitation and moisture flux on weekly averaged fields in 270 June-July at subseasonal time scale. Results are similar to seasonal time scale, where the moisture flux 271 has a higher predictability than precipitation at different lead times (Fig. 7). Moreover, the potential 272 predictability may change under different climatic conditions. The seasonal predictability is much 273 higher when initialized in warm ENSO conditions not only for precipitation but also for moisture flux. 274 More importantly, the moisture flux shows higher detectability (hit rate) than precipitation for extreme 275 pluvial flooding events following El Niño winters. The results suggest that it may be possible to extend 276 277 the predictability of Yangtze River summer floods and to provide more reliable early warning by using atmospheric moisture flux predictions. However, to which degree that moisture flux is connected with 278 precipitation and floods might be model dependent. It is necessary to explore their connections in a 279 multi-model framework (e.g., NMME; Kirtman et al., 2014; Shukla et al., 2016). 280

This study extends previous findings on the predictability of precipitation and moisture flux at synoptic 281 scales (Lavers et al., 2014) to seasonal time scales, and from atmospheric river-affected regions to the 282 East Asian summer monsoon region. Given that the transport of atmospheric moisture from oceanic 283 source regions is important for extreme rainfall in monsoon regions (Gimeno et al., 2012), moisture flux 284 285 might also be useful for long-range forecasting over other areas affected by the monsoon and low-level jets. In fact, extreme precipitation and floods are found to be associated with large-scale moisture 286 transport over the North American monsoon (Schmitz and Mullen, 1996) and the South American 287 monsoon (Carvalho et al., 20110) regions. Extreme precipitation and floods usually occur accompanied 288 with intensive atmospheric moisture transport, especially over a large area such as the middle and lower 289

290 reaches of the Yangtze River. Given higher predictability of atmospheric moisture flux, it can be used 291 as a precursor for flooding forecasting, either directly linking moisture flux to streamflow prediction 292 through statistical techniques (e.g., conditional distribution or Bayesian methods), or adding moisture 293 flux information into precipitation prediction, and consequently improving floods prediction. Moreover, 294 it is suggested that assimilating moisture flux observations into numerical climate forecast models 295 would benefit the prediction of hydrological extremes.

The higher moisture flux predictability largely arises from more predictable large-scale circulation (Li et al., 2016), which strongly determines the atmospheric moisture transport. Although precipitation variability is affected by both large-scale moisture transport and localized process and features, such as condensation nuclei in the atmosphere and lifting movement, it is expected that moisture transport could still be used as a crucial source of predictability for flooding over monsoonal regimes, especially at long leads where meso-scale convection is still unpredictable at seasonal time scales.

302

Acknowledgement. This work was supported by National Natural Science Foundation of China (91547103, 41605055), and the National Key R&D Program of China (2016YFA0600403). The authors thank Dr. Arun Kumar for helpful discussions. The authors acknowledge NCEP/EMC and IRI (http://iridl.ldeo.columbia.edu/SOURCES/.NOAA/.NCEP/.EMC/.CFSv2/) for making the CFSv2 hindcast and real-time forecast information available. We are also grateful for the constructive comments from three anonymous reviewers to improve the quality of this paper.

**310 References**

[revised manuscript text omitted]